# Physicochemical Characteristics of Soluble Dietary Fiber Obtained from Grapefruit Peel Insoluble Dietary Fiber and Its Effects on Blueberry Jam

**DOI:** 10.3390/foods11223735

**Published:** 2022-11-21

**Authors:** Jiayan Xie, Guanyi Peng, Xiaobo Hu, Jianhua Xie, Yi Chen, Ruihong Dong, Jingyu Si, Chaoran Yang, Qiang Yu

**Affiliations:** State Key Laboratory of Food Science and Technology, China-Canada Joint Laboratory of Food Science and Technology (Nanchang), Key Laboratory of Bioactive Polysaccharides of Jiangxi Province, Nanchang University, 235 Nanjing East Road, Nanchang 330047, China

**Keywords:** modification, monosaccharide compositions, color analysis, rheology, sensory qualities

## Abstract

Appropriate modification methods can increase the proportion of soluble dietary fiber (SDF). In this study, grapefruit peel insoluble dietary fiber (GP-IDF) was modified with the combined microwave and enzymatic method to obtain SDF. With regard to structural characterization, SDF from grapefruit peel IDF (GP-IDF-SDF) presented as a flat sheet with cracks, composed of a typical cellulose type I crystal, and had good stability below 200 °C. Galacturonic acid, arabinose and glucuronic acid were the main monosaccharide compositions, indicating that pectin might have been the principal component. Moreover, GP-IDF-SDF was excellent in water retention capacity (13.43 ± 1.19 g/g), oil retention capacity (22.10 ± 0.85 g/g) and glucose adsorption capacity (14.49 ± 0.068 mg/g). Thereafter, the effects of GP-IDF-SDF and commercial pectin addition on the color, rheology, texture and sensory properties of blueberry jam were compared. The results showed that the color of jam with GP-IDF-SDF was lighter. The addition of GP-IDF-SDF had less effects on the viscosity and gel strength of jam, but it enhanced the stability of jam. According to sensory data, the color, texture and spreadability of jam with GP-IDF-SDF or pectin were improved and more acceptable. Overall, GP-IDF-SDF had functional characteristics and played a positive role in jam, and it is expected to be a candidate for the development of functional food ingredients.

## 1. Introduction

Dietary fiber (DF) is abundant in vegetables, fruits, whole grains, legumes, etc. It is vital to human health and is known as the seventh nutrient in biology [1]. With the deepening of research, it has been found that the intake of DF might have lots of benefits for humans, such as alleviating diabetes and lowering blood pressure [2]. DF can be divided into two categories: insoluble dietary fiber (IDF), which contains cellulose, hemicellulose and lignin; and soluble dietary fiber (SDF), which contains pectin, gum, etc. SDF is more advantageous for its physiological activity due to its solubility; however, the content of SDF is lower than that of IDF in fruit [3]. Therefore, researchers have explored various methods to improve the IDF/SDF ratio in raw materials. For instance, the IDF/SDF ratio from okara dietary fiber was reduced after fermentation and microwave treatment, and the modified DF was identified as a potential functional food [4,5]. In previous research studies in our laboratory, the dietary fiber from rice bran, tea pomace and grapefruit peel was modified using microbial fermentation, the alkaline method and the combined microwave and enzymatic method, respectively [6,7,8]. It was found that the increase in the SDF yield was associated with the destruction of dietary fiber microstructure and increased functional properties. Of note, the total DF content was largely maintained, while the IDF content was decreased [8]. Therefore, one might speculate that the modification converted IDF to SDF, thus altering their proportion to enhance functionality. Meanwhile, the combined microwave and enzymatic method was proven to have a great impact on the structure of grapefruit peel insoluble dietary fiber (GP-IDF), and the destruction of the GP-IDF structure was found to contribute to the release of active substances [9]. However, physiological activities and applications of SDF converted from IDF have not been systematically elucidated.

Water retention capacity, gel properties, thickening, etc., are common properties of SDF, and SDF can be used in the food industry as a technical food component (stabilizer, thickener, gel, etc.) to improve food functionality [10]. Reportedly, SDF could significantly improve the stability of fluid foods, such as in the production of jelly, fermented milk or jam [11,12]. As an ancient and commonly used approach of fruit preservation, jam processing is very popular in European and American countries. Jam can be easily made by boiling the fruit puffer, sugar, water and citric acid together [13]. Nevertheless, in commercial production, a certain amount of pectin is often added to improve the water retention capacity of jam [14]. Compared with other thickeners, SDF not only has advantages in forming a network in jam to achieve the role of water retention but also has better physiological activity at a lower cost. Grigelmo-Miguel et al. found that there were no significant differences between strawberry jam with dietary fiber and commercial jam with pectin, and the sensory performance of jam was better [15].

To sum up, this study aimed to (1) explore the physical properties, water retention and oil retention, as well as glucose adsorption capacities, of SDF obtained from grapefruit peel IDF (GP-IDF-SDF); (2) investigate the effects of GP-IDF-SDF addition on the color, rheology, texture and sensory properties of conventional jam (with or without adding pectin); and (3) elucidate the potential application of GP-IDF-SDF. Our research can contribute to exploiting the value of IDF and can provide new ideas and a theoretical basis for the comprehensive utilization of dietary fiber.

## 2. Materials and Methods

### 2.1. Materials

Grapefruit peel was purchased from a local market in Ganzhou City, Jiangxi Province, China. All standard monosaccharaides, including mannose, glucose, arabinose, galactose, galacturonic acid, rhamnose, fructose and xylose, were purchased from Shanghai YuanYe Biotechnology Co., Ltd. (Shanghai, China). Dextrans of various molecular weights were purchased from Aladdin Biotechnology Co., Ltd. (Shanghai, China). Low-methoxy pectin, sucrose and citric acid were purchased from Wanbang industrial Co., Ltd. (Zhengzhou, China). The other regents used in this experiment were of analytical grade.

### 2.2. Preparation of SDF from Grapefruit Peel Insoluble Dietary Fiber (GP-IDF-SDF)

GP-IDF was processed according to the method described in a previous study [9]. Briefly, GP-IDF was modified with microwave treatment (600 W, 85 °C, 37 min), followed by enzymatic treatment (addition at 0.8%, temperature of 60 °C, incubated for 2 h). Thereafter, the mixture was centrifuged at 4800 rpm for 15 min, during which the supernatant was added four times in volumes of 95% ethanol and left to stand for 24 h. Subsequently, the precipitate was re-dissolved with distilled water; the organic phase and part of the water in the solution were removed via rotary evaporation, and the rest of the solution was freeze-dried (Freeze dryer; Labconco, Kansas City, MO, USA) to obtain GP-IDF-SDF.

### 2.3. Monosaccharide Composition

The monosaccharide composition of GP-IDF-SDF was detected with high-performance anion-exchange chromatography (HPAEC; Dionex ICS-5000; Thermo Fisher Scientific, Waltham, MA, USA); then, data were collected and processed using Chromeleon 6.8 (Thermo Fisher Scientific, Waltham, MA, USA) software. Before the determination of monosaccharide composition, GP-IDF-SDF was subjected to a series of treatments based on the literature reported by Du et al. [16]. The sample (5 mg) was mixed with 0.5 mL of 12 M H_2_SO_4_ in an ice bath for 30 min; then, 2.5 mL of ultra-pure water was added and incubated in a silicone oil bath for another 3 h. Next, the mixture was diluted again to a concentration of 20 μg/mL and filtered through a 0.22 μm syringe filter.

### 2.4. Molecular Weight

The molecular weight of GP-IDF-SDF was determined with the high-performance gel permeation chromatography (HPGPC) method using an Agilent 1260 HPLC system (Agilent, Santa Clara, CA, USA) with an Ultrahydrogel™-1000 column (300 mm × 7.8 mm) and a 2414 RI refractive index detector [17]. The standard curve was produced using Dextran T standards (Mw: 10 kDa, 40 kDa, 70 kDa, 500 kDa and 2000 kDa). GP-IDF-SDF and dextran standards (1 mg/mL) were dissolved in ultra-pure water containing 0.02% NaN_3_. Filtration through a 0.22 μm membrane filter was performed before injecting the solution into the chromatographic system.

### 2.5. Structural Characterization of GP-IDF-SDF

#### 2.5.1. Scanning Electron Microscopy (SEM)

The microscopic images of GP-IDF-SDF were observed with a scanning electron microscope (JSM 6701F; JEOL Ltd., Tokyo, Japan) and collected at an accelerating voltage of 10 kV and at 100× and 500× magnifications [6].

#### 2.5.2. X-ray Diffraction (XRD)

The XRD measurement of GP-IDF-SDF was obtained using a diffractometer (D8 Advance; Bruker, Saarbrucken, Germany) within the scanning range of 5–50°. Other conditions were as follows: radiation voltage, 30 kV; current, 20 mA. The relative crystallinity of the sample was calculated by referring to the Segal method [18].

#### 2.5.3. Thermal Properties

Referring to the study by Xie et al. [19], a thermogravimetric analyzer (TGA 4000; PerkinElmer, Waltham, MA, USA) was used to test the thermal properties of GP-IDF-SDF. Before the experiment, the internal gas environment was purged with high-purity nitrogen at a flow rate of 20.0 mL/min. Thereafter, under a nitrogen atmosphere, thermogravimetric analyses were performed at a heating rate of 10 °C/min in the temperature range of 30–600 °C.

#### 2.5.4. Fourier Transfer-Infrared Spectroscopy (FT-IR)

FT-IR (FTIR; Nicolet, Madison, WI, USA) was used to detect the functional groups of GP-IDF-SDF [8]. Mixtures of sample and KBr (1:100, *w*/*w*) were ground in an agate mortar. Spectra were read over the range of 4000 to 400 cm^−1^ with a resolution of 32 scans and 4 cm^−1^.

### 2.6. Hydration Properties of GP-IDF-SDF

#### 2.6.1. Water Holding Capacity (WHC)

About 1.0 g of sample was accurately weighed and placed into a centrifuge tube; then, 30 mL of distilled water was added. After soaking for 1 h at room temperature, the mixture was centrifuged at 4800 rpm for 15 min to remove the supernatant [20]. Thereafter, the residue was drained and weighed. The WHC was calculated by referring to the following Equation (1):WHC(g/g) = (W_2_ − W_1_)/W_1_(1)
where W_1_ is the weight of SDF and W_2_ is the weight of the sample soaked in water.

#### 2.6.2. Oil Holding Capacity (OHC)

About 1.0 g of sample was accurately weighed and placed into a 50 mL centrifuge tube; then, 30 mL of distilled soybean was added [20]. After soaking for 1 h at room temperature, the mixture was centrifuged at 4800 rpm for 15 min to remove the excess oil. The OHC was calculated by referring to the following Equation (2):OHC(g/g) = (W_2_ − W_1_)/W_1_(2)
where W_1_ is the weight of the sample and W_2_ is the weight of the sample soaked in oil.

#### 2.6.3. Glucose Adsorption Capacity (GAC)

The GAC of GP-IDF-SDF was investigated according to a previously described method with slight modifications [21]. About 0.1 g of sample was added to 5 mL glucose solution (0.5 mg/mL). The mixture was stirred at room temperature for 1 h and then centrifuged at 4800 r/min for 12 min. Subsequently, 2 mL of supernatant was mixed with 1.5 mL dinitro-salicylate (DNS) before being placed into boiling water for 5 min. Upon reaching room temperature, the mixture was mixed evenly with 12.5 mL of distilled water and measured at 540 nm. The residual concentration of glucose was calculated based on the standard curve (y = 0.98807x − 0.00205, R^2^ = 0.9954).
GAC (mg/g) = (M_1_ − M_2_)/W_1_(3)
where W_1_ is the weight of GP-IDF-SDF, and M_1_ and M_2_ represent the weight of glucose in the solution before and after adsorption, respectively.

### 2.7. Formulation of Blueberry Jam

A portion of sucrose (60% of total sucrose) was added into blueberry pulp; then, the mixture was stirred and cooked at 90 °C for 30 min until the soluble solid content reached 28.9%. Subsequently, the rest of sucrose, citric acid and GP-IDF-SDF or low-methoxy (LM) pectin were added into the pulp, which we continued to cook until the soluble solid content reached 40%. Thereafter, the jam was immediately poured into a sealed glass can, followed by sterilization in a boiling water bath for 10 min. The detailed recipe of jam is shown in Appendix A.

### 2.8. Color Measurement

The color of jam was assessed with a Color-Flex 45/0 spectrophotometer (Hunter Associates Laboratory Inc., Reston, VA, USA). CIE-L*a*b*color parameters were recorded as L* (lightness), a* (redness/greenness) and b* (yellowness/blueness) values [22]. Four replicates were carried out for each sample. Chrome (C*) and the total color difference (Δ*E*) were calculated according to Equations (4) and (5), respectively.
(4)$$C^{*}=\sqrt{{a*}^{2}{+b*}^{2}}$$
(5)$$\Delta E=\sqrt{\Delta {L*}^{2}+\Delta {a*}^{2}+\Delta {b*}^{2}}$$

### 2.9. Rheological Properties of Jam

The rheological properties of blueberry jam were tested with an ARES-G2 rheometer (TA Instruments, New Castle, DE, USA) at 25 °C with a 1 mm gap between the parallel plate (40 mm) and the objective table [23]. Samples were equilibrated for 2 min before testing.

#### 2.9.1. Flow Sweep

Flow curve measurements were carried out by increasing the shear rate from 0.01 to 300 s^−1^, and the relationships among viscosity, shear stress and shear rate were recorded [24]. The data points were regressed using the Herschel–Bulkley model, and the fitting precision was expressed with R-square. The equation is
(6)$$\sigma =K{\left(\gamma \right)}^{n}+\sigma_{0}$$
where σ is the shear stress (Pa); K is the consistency index; γ is the shear rate; *n* is the flow behavior index; σ_0_ is the yield stress (Pa).

For the determination of the thixotropy of jam, we set the shear rate to increase from 0.01 to 300 s^−1^ and then decrease from 300 to 0.01 s^−1^, and the relationships between shear stress and shear rate were recorded.

#### 2.9.2. Oscillatory Frequency Sweep

The linear viscoelastic region was measured to determine the scanning strain value with amplitude sweep before oscillatory frequency sweep. The storage modulus (G′) and loss modulus (G″) of the samples were measured at frequencies of 0.1–10.0 Hz and a strain of 1.0%.

### 2.10. Texture Characteristics of Jam

Texture properties including gel strength, hardness, consistency and adhesiveness of blueberry jam were examined using a texture analyzer (TA-XT plus; Stable Co., Surrey, UK) equipped with a 0.5 R cylinder probe. Referring to the previous literature with slight modifications [22], the jam was transferred into plastic cups, with a 36 mm diameter and a 50 mm height (sample height, 40 mm), and the parameters were set as follows: pre-test speed, 5.0 mm/s; test speed, 1.0 mm/s, post-test speed, 5.0 mm/s; trigger detection force, 5 g; penetration distance, 15 mm.

### 2.11. Sensory Evaluation

Sensory evaluations were made on the second day after the jam was made, and the jam samples were numbered before the tests. Thirty panelists rated the color, odor, texture, taste and overall acceptability of the jam samples on a scale of 5, from 1 (significantly dislike) to 5 (significantly like) [25]. The sensory inspectors were provided with slices of bread and spoons for evaluating spreadability, as well as drinking water for gargling. Finally, the average sensory score of the group members was analyzed.

### 2.12. Statistical Analysis

The above experiments were repeated at least three times, and the results were analyzed using an ANOVA performed with SPSS software (version 21.0, SPSS Inc., Chicago, IL, USA) and were expressed as means ± standard deviation (S.D.). The confidence level of statistical significance was set to the probability value of 0.05.

## 3. Results and Discussion

### 3.1. Monosaccharide Composition and Molecular Weight of GP-IDF-SDF

The yield of GP-IDF-SDF was 9.2 ± 0.36%. A preliminary study in our laboratory found that SDF content in grapefruit peel was about 3.62 ± 0.13%, and the yield of SDF could be significantly increased via microwave modification (7.94 ± 0.20%). Considering that the total dietary fiber content was maintained at a similar level, we hypothesized that microwave treatment disrupted the structure of IDF, thus inducing the conversion of IDF to SDF [8]. In addition, we also found that the combined microwave and enzymatic treatment of GP-IDF could release polyphenols effectively [9]. These findings confirmed that the modification process of dietary fiber might be accompanied by the release of some soluble components, and combined microwave and enzymatic treatment might be a new, effective way to improve the utilization of IDF.

According to the results of the monosaccharide composition shown in Appendix A, arabinose (Ara), galactose (Gal), glucose (Glu), xylose (Xyl), mannose (Man) and galacturonic acid (Gala) were typical components of GP-IDF-SDF, which is similar to the composition of SDF in grapefruit peel [8]. Interestingly, Man, a monosaccharide that was not previously detectable in grapefruit peel soluble dietary fiber, was detected in GP-IDF-SDF, which was consistent with the results found in modified okara dietary fiber [6]. The structure of IDF was disrupted after microwave and enzyme treatment, such as glycosidic bonds and intermolecular interactions. As a consequence, it could be speculated that hemicellulose in grapefruit peel was able to be hydrolyzed into Xyl, Man, Ara, etc., after combined microwave and enzymatic modification. Additionally, fructose might have been converted to glucose during this process, since it was no longer detectable in GP-IDF-SDF. Gal, Ara and Gala were the typical compositions of pectin [26], and the molar ratio of monosaccharides in GP-IDF-SDF was 3.86 (Ara):3.04 (Glu):2.19 (Xyl):1.11 (Man):6.06 (Gala):1 (Gal). It was found that Gala accounted for the largest proportion in GP-IDF-SDF, revealing that pectin might have been the major component of GP-IDF-SDF. In light of these analyses, we hypothesized that the SDF prepared with the combined microwave and enzymatic modification might have gel properties similar to those of pectin.

The molecular weight (Mw) of GP-IDF-SDF was analyzed with HPGPC, and multiple peaks were observed after elution (Figure 1), indicating that GP-IDF-SDF was a heterogeneous polymer [6]. The calibration equation (y = −0.2004x + 1.4494) derived from the linear regression of the calibration curve was used to calculate the average Mw. A broad peak, a shoulder peak and a narrow peak were observed at 14.293 min, 17.37 min and 18.84 min, and the corresponding Mw values were 167.40 kDa, 1.51 kDa and 0.15 kDa, respectively. It was noted that the peak area was correlated to the proportion of each component in the sample, so the Mw of the component with the highest proportion in GP-IDF-SDF was 167.40 kDa. The average Mw of SDF in grapefruit peel was reported to be in the range of 0.16–620 kDa [8], suggesting that IDF in grapefruit peel was converted to SDF with low molecular weight during combined microwave and enzymatic modification.

### 3.2. Structural Characterization of GP-IDF-SDF

#### 3.2.1. Microstructure Properties

Different types and sources of dietary fiber reveal their own unique microstructures. The observed microstructure of GP-IDF-SDF was a flat sheet with cracks (Figure 2). A previous study found that SDF extracted from grapefruit peel via microwave treatment was a sheet-like structure with holes [8]; this particular structure facilitates an increase in the relative surface area of dietary fiber, thus exhibiting advantageous functional properties, such as WHC, OHC, GAC, etc. Therefore, it could be hypothesized that GP-IDF-SDF with cracks might be able to contribute favorably to the functional properties of dietary fibers.

#### 3.2.2. X-ray Diffraction and Thermal Properties

According to XRD data (Figure 3a), GP-IDF-SDF showed a weak diffraction peak at 12.13° and a strong diffraction peak at 21.53°, which was similar to the previous results and indicated that there was a crystallization region (typical cellulose I crystal) in GP-IDF-SDF [19]. However, the “bun-shaped” curve revealed the presence of amorphous features [6]. GP-IDF-SDF exhibited the typical diffraction pattern characteristics of polymers, which was attributed to the fact that SDF is a mixed substance with a crystal region (e.g., cellulose) and an amorphous region (e.g., hemicellulose).

Thermal stability is an important characteristic of polymers that can be analyzed with a TGA. As depicted in Figure 3b, GP-IDF-SDF displayed a multi-step thermal degradation process. The first stage (30–150 °C) was mainly due to the evaporation of free water and bound water in SDF [19], at which the weight loss of GP-IDF-SDF slowly decreased by about 7.9%. In the range of 150–200 °C, weight loss was still slow, as the macromolecular substances were depolymerized gradually [6]. Subsequently (200–400 °C), the weight of GP-IDF-SDF decreased rapidly, which was attributed to the pyrolysis of hydrogen bonds, glycosidic bonds or carbon chains in polysaccharides [27]. With the pyrolysis temperature rise to the last stage (500–600 °C), the weight loss slowed as carbon thermally decomposed, and the final residual mass was close to 23.0%. In this case, the processing temperature should not exceed 200 °C to avoid thermal decomposition of GP-IDF-SDF, which might lead to the loss of physicochemical and functional properties.

#### 3.2.3. FT-IR Analysis

To explain the functional groups of GP-IDF-SDF, FT-IR spectra were obtained (Figure 4). Similar to previous results of soluble dietary fibers, there was a strong and wide absorption peak at about 3000–3650 cm^−1^, and it was ascribed to the O-H vibration stretching of hydrogen and hydroxyl groups [7]. The peak at 2924 cm^−1^ was indicative of the stretching vibration of C-H, including the bending vibrations of CH, CH_2_ and CH_3_, which are typical structures of polysaccharide compounds [6]. Particularly, the peaks at 1740 cm^−1^ and 1629 cm^−1^ represented the stretching vibrations of the esterified carboxyl group (-COOR) and free carboxyl group (-COOH), respectively [28]. The peaks at 1018–1300 cm^−1^ were ascribed to C-O stretching, which might be the vibration of C-O-H and C-O-C in the sugar ring or the vibration of primary alcohols [5]. Peaks around 973 cm^−1^ and 1018 cm^−1^ were recognized as the characteristic absorption peaks of pyranose, indicating that the configuration of dietary fiber was linked by pyran glycosidic bonds [29]. Additionally, a weak peak at around 917 cm^−1^ was assigned to β-glycosidic linkages present in hemicellulose [30]. Altogether, the above results clearly demonstrated that GP-IDF-SDF had the typical functional groups of polysaccharides, and these groups played an essential role in the functional properties of SDF, such as water retention, oil retention and glucose adsorption capacities.

### 3.3. Functional Properties

WHC, GAC and OHC are important functional properties of SDF. As reported in the literature, WHC can positively affect the viscosity of food and can prevent food from dehydrating as well as contracting [8,31]. The GAC and OHC of SDF endow food with higher edible value, such as delaying the rise in postprandial blood glucose levels, absorbing oil in food and reducing the amount of dietary fat intake [32]. GP-IDF-SDF exhibited good hydration abilities, with WHC and OHC values of 13.43 ± 1.19 g/g and 22.10 ± 0.85 g/g, respectively, which were greater than the values reported for orange dietary fiber [33]. Consistent with the results of SEM and FTIR analyses, the structure with cracks exposed a considerable number of hydrophilic and hydrophobic groups in dietary fibers, thus providing the basis for their physicochemical properties. Dietary fiber with high OHC could be used as an ingredient to stabilize foods with high emulsion content. Additionally, with regard to the food industry, dietary fiber with high WHC could be used in food systems that need to absorb or retain water, such as jams, jellies, baked foods, etc., to improve their performance, extend shelf life and reduce production costs. For instance, dietary fibers from apple, bamboo, psyllium and wheat were added to jams to provide the desired functional properties [12]. Moreover, SDF is capable of absorbing glucose via the surface groups; our study found that the GAC value was 14.49 ± 0.068 mg/g, indicating that GP-IDF-SDF might have hypoglycemic effects in vitro. On the whole, SDF extracted from GP-IDF might have good functional properties; as a consequence, it was added to jam to further investigate its feasibility for incorporation into food products.

### 3.4. Color Measurement of Jam

In order to study the effects of SDF on jam qualities, four groups were set as follows: jam prepared with 1% and 0.5% GP-IDF-SDF (GPSJ-1 and GPSJ-2), jam prepared with LM pectin (LMJ) and base jam without the addition of pectin or dietary fiber (BJ).

The color of a food system is not only associated with the physical state and chemical composition of the food, but it is also related to the structure [34]. The L * (light) values of LMJ, GPSJ-1 and GPSJ-2 were significantly higher than that of BJ, reflecting that the lightness of blueberry jam was enhanced by the addition of dietary fiber or pectin. One of the reasons might be that dietary fiber and pectin could attenuate the Maillard reaction during jam processing, thus significantly improving the reflectance and brightness of jam [35]. Furthermore, LMJ and GPSJ-2 were darker (lower L* values) than GPSJ-1, implying that GP-IDF-SDF was superior to LM pectin in maintaining the brightness of jam. From the above, it was speculated that the effect of GP-IDF-SDF on attenuating the Maillard reaction was more prominent than that of LM pectin during jam processing.

It was reported that LM pectin could increase the a* value [22], and this was in line with our results. When LM pectin was replaced by GP-IDF-SDF, the a* value decreased from 2.02 ± 0.13 (LMJ) to 1.08 ± 0.11 (GPSJ-1) and to 1.55 ± 0.13 (GPSJ-2), respectively. Moreover, the a* value of GPSJ-1 was close to that of BJ, revealing that GP-IDF-SDF had no obvious effects on the redness of jam. Additionally, the incorporation of GP-IDF-SDF increased the b* value, indicating that GPSJ-1 was yellower than BJ. From the above results, as compared with basic jam, the addition of GP-IDF-SDF resulted in lighter-colored jam.

The C* (chrome) value and Δ*E* (chromatic aberration) were calculated to characterize the variation in the color parameters (Table 1). GP-IDF-SDF and LM pectin could both retain the C* value of jam, reflecting the fact that the jam samples were bright (GPSJ-1 and LMJ). Taking BJ as the standard reference sample, Δ*E* (chromatic aberration) was calculated with reference to equation (6). Δ*E* showed significant differences among LMJ (2.54 ± 0.27), GPSJ-1 (3.85 ± 0.05) and GPSJ-2 (3.34 ± 0.25). Interestingly, the Δ*E* values of GPSJ-1 and GPSJ-2 were both relatively higher and significantly greater than that of LMJ, suggesting that the presence of GP-IDF-SDF led to a significant difference in the color of the jam samples. Igual et al. reported that the color of grapefruit jam supplemented with bamboo fiber had a higher Δ*E* value (compared with jam without dietary fiber), while there were no significant differences in the Δ*E* value compared with the color of fresh fruit, indicating that jam supplemented with fiber might be close to fresh fruit puree [35].

### 3.5. Rheological and Textural Properties of Jam

The variations in the shear stress and viscosity according to the shear rate are illustrated in Figure 5. With the increase in the shear rate, the shear stress increased, and the viscosity decreased, suggesting that the jam was a shear-thinning fluid, which was in accordance with the Herschel–Bulkley model. The yield stress (σ0), flow index (*n*) and consistency coefficient K were obtained, and the R-square values were more than 0.99.

As shown in Table 2, the yield stress values of jam samples supplemented with LM pectin were higher than those of the other jams, reflecting that they had better resistance to external force. Furthermore, the flow index (*n*) values of four jam samples were less than 1, which indicated that the jam samples were non-Newtonian pseudoplastic fluids. It was reported that the smaller the flow index is, the more obvious the shear-thinning characteristics are. As a consequence, LMJ and GPSJ-2 exhibited greater pseudoplasticity than the other jams. The consistency index (K) is related to the viscosity characteristics of fluid. A greater the consistency index means a greater viscosity and stronger gel properties of the jam. Therefore, jam supplemented with LM pectin and GP-IDF-SDF exhibited higher viscosity and gel characteristics than basic jam (BJ), which might be attributed to their strong water retention capacities [23]. The consistency coefficient of LMJ was found to be higher than that of GPSJ-1, followed by GPSJ-2, indicating that although both LM pectin and GP-IDF-SDF could improve the gel characteristics of jam, GP-IDF-SDF was not as effective as commercial LM pectin, and this could be caused by their structural inconsistency.

In terms of the thixotropy of jam (Figure 5c), the area (hysteresis loop area) was enclosed by two ascending and descending lines, and it was related to the structural damage of jam during shearing [35]. The value of the hysteresis area of these jam samples are listed in Table 2. Interestingly, the hysteresis loop area of jam fully supplemented with GP-IDF-SDF (GPSJ-1) was significantly smaller than that of BJ, which suggested that GPSJ-1 had less thixotropy, indicating that it was more susceptible to retain the original shape and more capable of avoiding precipitation and hanging phenomena during production [23]. Additionally, LM-pectin-supplemented jam was reported to have a larger hysteresis loop area, leading to greater energy dissipation, since particles with poor fluidity entrained in the pectin network cannot flow directionally [24]. This might have been the reason why the hysteresis loop area of LMJ was the largest. The hysteresis loop area of GPSJ-2 was significantly smaller than that of LMJ, revealing that GP-IDF-SDF might improve the pectin network, so that the particles in the network could flow with shearing. Briefly, the addition of GP-IDF-SDF could probably improve the stability of jam during production and reduce the energy consumption.

The variations in the storage modulus (G′) and loss modulus (G″) according to the angular frequency were analyzed (Figure 5d). G′ and G″ are related to the elasticity and viscosity of the system, respectively. During oscillatory frequency sweep, the G′ values of all kinds of jam were greater than the G″ values, indicating that the elasticity of jam was more obvious than viscosity; that is, these jam samples showed weak gel characteristics or a primarily solid behavior. Weak gels are capable of recovering after being deformed by external forces and thus are stable for long-term storage [12]. In addition, the G′ values of LMJ, GPSJ-1 and GPSJ-2 were higher than that of BJ, indicating that the solid properties of jam were strengthened with LM pectin or GP-IDF-SDF addition. In the high-angular-frequency range, the G′ values of GPSJ-2 and LMJ were closer and were larger than those of GPSJ-1 and BJ, suggesting that the addition of LM pectin could obviously enhance the solid properties of jam, which resulted in stronger hardness and toughness of LMJ and GPSJ-2. Of note, excessive hardness leads to poor spreadability [12]. Therefore, GP-IDF-SDF and LM pectin had a certain similarity in improving the solid properties and elasticity of jam, but jam with GP-IDF-SDF was relatively softer.

A texture analysis can be considered as an imitation of the mastication operation; the texture properties of blueberry jam are shown in Table 2. The results of consistency and gel strength were consistent with the above rheological results, indicating that the incorporation of LM pectin and dietary fiber in blueberry jam seemed to favor the entanglement of the gel network, thus improving the stability of jam during production. Additionally, hardness and adhesiveness are important indicators of jam spreadability and ductility, measuring the resistance of jam to deformation under an applied force and the effort required to remove the probe from jam, respectively [22]. Both of them were increased after LM pectin and GP-IDF-SDF addition, especially in LMJ. However, the higher hardness of LMJ and GSPJ-2 led to a lower malleability than that of BJ and GSPJ-1, which could have been due to the differences in physicochemical properties between pectin and SDF.

Taken together, GP-IDF-SDF has the potential to become a new food additive with gel functions in the jam industry. GP-IDF-SDF and LM could enhance the gel properties and viscosity, and the latter was more effective than the former. However, the texture of LMJ was hard, and as a consequence, it was not favorable for its malleability.

### 3.6. Sensory Evaluation

The results of the sensory evaluation are shown in a radar map (Figure 6). The odor and taste of GPSJ-1, GPSJ-2 and LMJ were analogous to those of BJ, indicating that the addition of LM pectin and GP-IDF-SDF had little effect on these parameters of blueberry jam. Notably, the color parameters of LMJ, GPSJ-1 and GPSJ-2 received higher scores, especially GPSJ-1, which was similar to the result of the L* value previously obtained with spectroscopy, that is, the addition of dietary fiber and LM pectin contributed to brighten and naturalize the blueberry jam samples. In addition, it was found that LMJ, GPSJ-1 and GPSJ-2 had similar overall acceptability scores, which might have been related to their excellent performance in terms of texture and spreadability (important indicators for testing jam quality). This could have been attributed to their good viscosity and gel strength. Consequently, GP-IDF-SDF was comparable to commercial LM pectin, and its incorporation into blueberry jam might make jam more palatable or could improve the edible quality.

## 4. Conclusions

In this study, the physicochemical properties of SDF obtained from the modification of IDF from grapefruit peel were investigated for the first time, and its role in the production of blueberry jam was explored. The findings revealed that GP-IDF-SDF exhibited great abilities of water retention, oil retention and glucose adsorption. It might be used as a good thickener and stabilizer in the production of blueberry jam owing to its good properties. Compared with jam with LM pectin addition, jam supplemented with GP-IDF-SDF was brighter and had relatively weak gel properties, which might have been attributed to the differences in the structure and physicochemical properties of pectin and dietary fiber. In the sensory evaluation, jam supplemented with GP-IDF-SDF, similar to jam supplemented with LM pectin, had better spreadability and could improve the edible quality of jam, indicating that GP-IDF-SDF might be a new economical and powerful additive in the field of food processing. Therefore, the further optimization of the dietary fiber content and the preparation of jams with good performance in various aspects are necessary in the next step of research.

## Figures and Tables

**Figure 1 foods-11-03735-f001:**
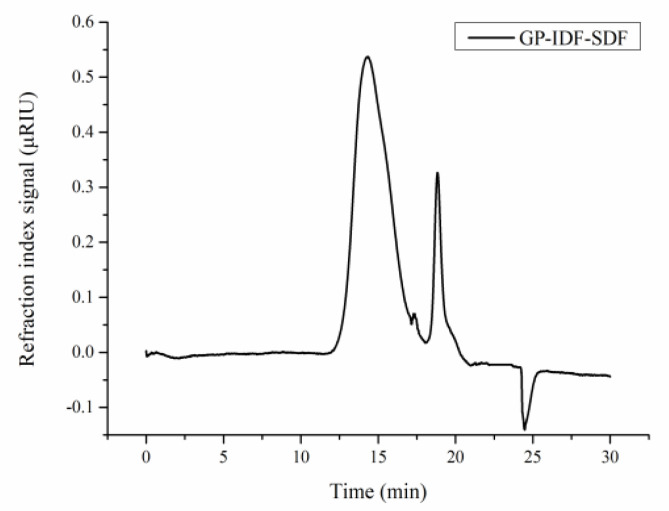
Gel permeation chromatogram profile of GP-IDF-SDF.

**Figure 2 foods-11-03735-f002:**
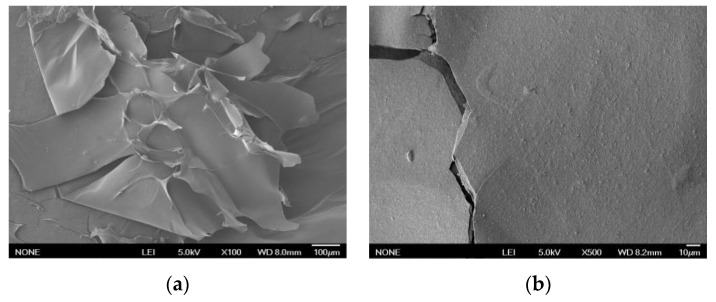
Scanning electron micrographs of GP-IDF-SDF collected at 100× (**a**) and 500× (**b**) magnifications.

**Figure 3 foods-11-03735-f003:**
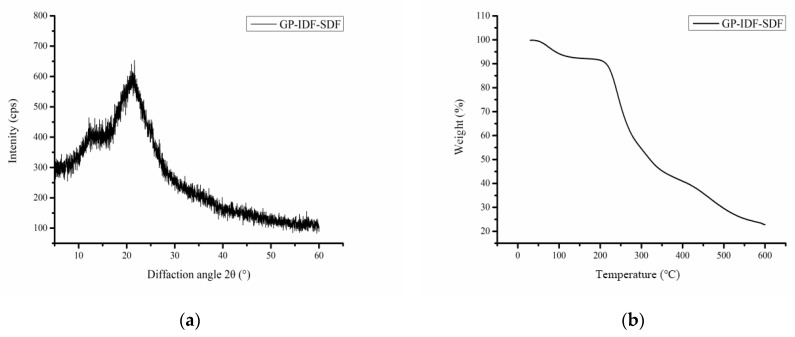
X-ray diffraction patterns (**a**) and thermal properties (**b**) of GP-IDF-SDF.

**Figure 4 foods-11-03735-f004:**
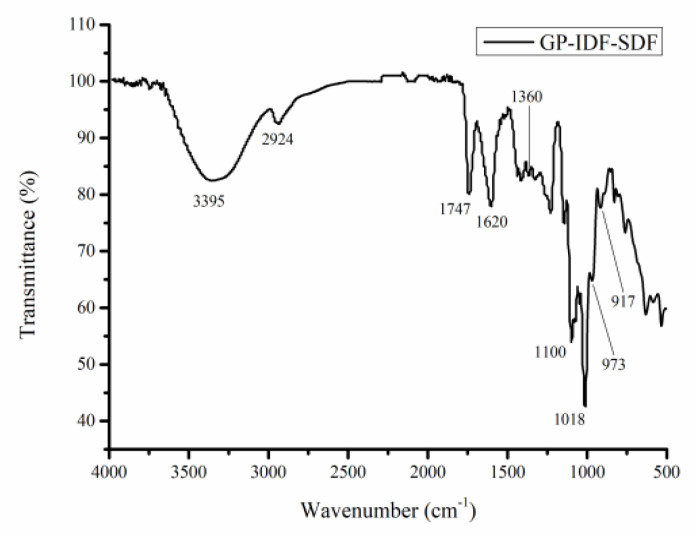
FT-IR spectrum (c) of GP-IDF-SDF.

**Figure 5 foods-11-03735-f005:**
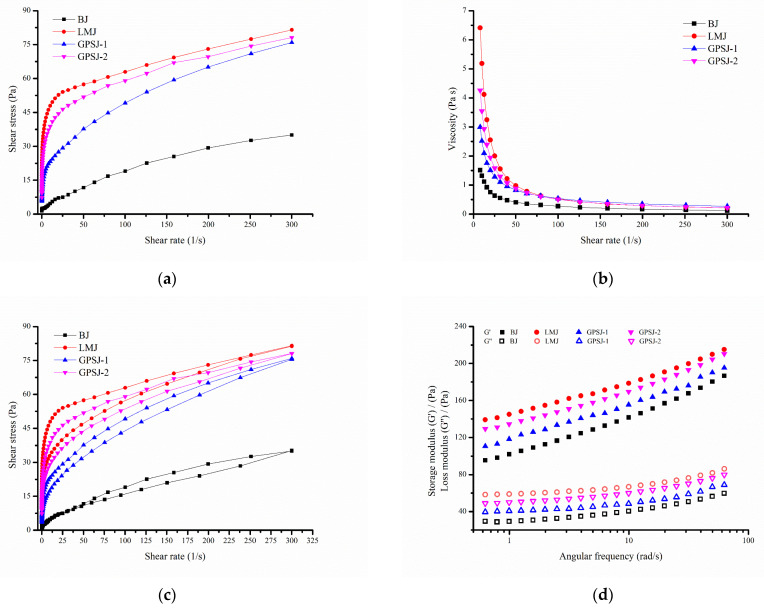
The rheological characteristics of blueberry jam. Variations in shear stress (**a**) and viscosity (**b**) with the shear rate. (**c**) Thixotropy. (**d**) Variations in storage modulus (G′) and loss modulus (G″) with the angular frequency.

**Figure 6 foods-11-03735-f006:**
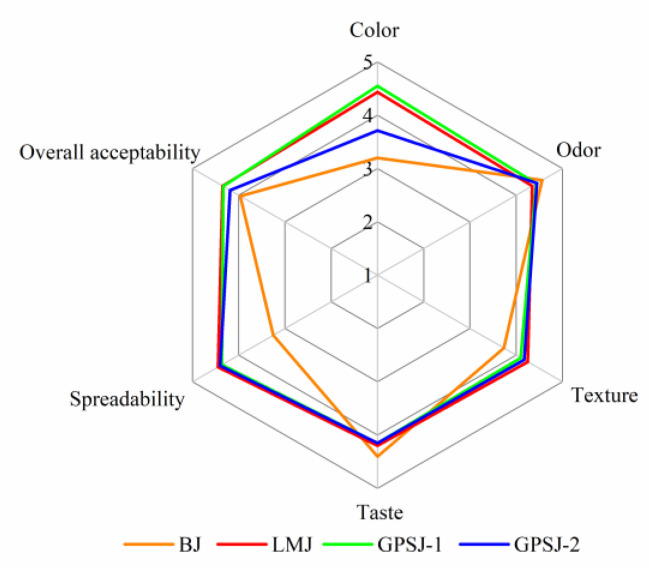
Sensory evaluation of blueberry jam samples.

**Table 1 foods-11-03735-t001:** Color parameters of blueberry jams.

Sample	L*	a*	b*	c*	Δ*E*
BJ	20.72 ± 0.17 c	1.34 ± 0.11 bc	2.18 ± 0.06 b	2.56 ± 0.05 b	—
LMJ	22.52 ± 0.62 b	2.02 ± 0.26 a	2.71 ± 0.77 ab	3.42 ± 0.59 a	2.54 ± 0.27 c
GPSJ-1	24.22 ± 0.08 a	1.08 ± 0.11 c	3.11 ± 0.07 a	3.30 ± 0.07 a	3.85 ± 0.05 a
GPSJ-2	23.24 ± 0.55 b	1.55 ± 0.13 b	2.31 ± 0.45 ab	2.79 ± 0.40 ab	3.34 ± 0.25 b

BJ, basic jam prepared with blueberry pulp, sucrose and citric acid; LMJ, jam prepared with blueberry pulp, sucrose, citric acid and 1% LM pectin; GPSJ-1, jam prepared with 1% GP-IDF-SDF; GPSJ-2, jam prepared with 0.5% GP-IDF-SDF and 0.5% LM pectin. Significant differences (*p* < 0.05) in the same column are expressed using different letters. “—” implies that no comparison was needed.

**Table 2 foods-11-03735-t002:** Herschel–Bulkley and textural parameters for blueberry jam.

	Parameter	BJ	LMJ	GPSJ-1	GPSJ-2
Rheological determination	Yield stress (g)	1.52 ± 0.13 c	10.00 ± 3.21 a	4.43 ± 0.34 b	8.76 ± 0.87 a
	Consistency index, K (Pa s^n^)	0.77 ± 0.059 d	42.04 ± 3.38 a	7.19 ± 0.32 c	26.62 ± 0.94 b
	Flow behavior index, *n* (-)	0.67 ± 0.014 a	0.13 ± 0.0085 c	0.40 ± 0.0048 b	0.19 ± 0.0048 c
	R-square	0.99585	0.99732	0.99446	0.99838
	Hysteresis loop area, S (Pa/s)	1357.63 ± 2.31 b	1448.91 ± 5.71 a	1245.70 ± 15.01 c	1375.57 ± 25.48 b
Texture determination	Gel strength (g)	5.26 ± 0.18 d	10.47 ± 0.30 a	6.73 ± 0.26 c	8.62 ± 0.66 b
	Hardness (g)	7.24 ± 0.31 c	15.05 ± 0.43 a	8.64 ± 0.45 c	11.96 ± 0.85 b
	Consistency (g*sec)	61.51 ± 1.76 c	112.35 ± 3.40 a	70.06 ± 2.72 c	90.69 ± 2.94 b
	Adhesiveness (g)	5.56 ± 0.22 c	18.45 ± 0.67 a	11.08 ± 1.85 b	15.91 ± 1.19 a

BJ, basic jam prepared with blueberry pulp, sucrose and citric acid; LMJ, jam prepared with blueberry pulp, sucrose, citric acid and 1% LM pectin; GPSJ-1, jam prepared with 1% GP-IDF-SDF; GPSJ-2, jam prepared with 0.5% GP-IDF-SDF and 0.5% LM pectin. Significant differences (*p* < 0.05) in the same column are expressed using different letters.

## Data Availability

Data are contained within the article or Appendix A.

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
