# Peer review of "Physicochemical Characteristics of Soluble Dietary Fiber Obtained from Grapefruit Peel Insoluble Dietary Fiber and Its Effects on Blueberry Jam"

_foods, 2022, doi:10.3390/foods11223735_

Round 1
Reviewer 1 Report
The manuscript describes the influence of grapefruit dietary fibers on the physicochemical characteristics of blueberry jam. The aim of this study is not clear, the language is poor and the results are poorly discussed
Other comments:
The title is not appropriate. Please change the title
Line 12: to obtain GP-IDF-SDF
Line 12: what is “GP-IDF-SDF”?
Line 160: Color of jams was …
Line 182: Which textural test was performed? (Add more details about texture measurement)
Line 190: Sensory evaluation is not correct. Why you have used 10 as the highest score? It should be an odd number (9, 7 or 5)
Line 242: filamentous structure?! It is a flat sheet with a crack.
Figure 1: Please explain what are the right hand and the left hand images.
Table 1: The caption and the treatments are not written
Sensory analysis and color parameters should be added to the manuscript (not in the supplementary file)
Author Response
General Comments: The manuscript describes the influence of grapefruit dietary fibers on the physicochemical characteristics of blueberry jam. The aim of this study is not clear, the language is poor and the results are poorly discussed.
Response to comment: Thank you for thorough evaluation of the manuscript and for your valuable advice. The comments you raised have been considered carefully and revisions are amended in the revised manuscript.
Point 1: The title is not appropriate. Please change the title.
Response 1: Thank you very much for the constructive suggestions. In order to highlight our research content, we have modified the title of the manuscript. The revised title is: Physicochemical characteristics of soluble dietary fiber from grapefruit insoluble dietary fiber and its effects on blueberry jam.
Point 2: Line 12: to obtain GP-IDF-SDF.
Response 2: We were very sorry for our carelessness, and the mistake has been corrected in the revised manuscript. Please refer to Line 12 marked in red.
Point 3: Line 12: what is “GP-IDF-SDF”?
Response 3: We are very sorry for our carelessness. The full name of “GP-IDF-SDF” has been added in the revised manuscript with red fonts. Please refer to Line 12 marked in red.
Point 4: Line 160: Color of jams was …
Response 4: Thank you very much for the constructive suggestions. The mistake has been corrected in the revised manuscript. Please refer to Line 161 marked in red.
Point 5: Line 182: Which textural test was performed? (Add more details about texture measurement).
Response 5: Thank you very much for the constructive suggestions. We have supplemented the details about texture measurement. Please refer to Lines 185-191.
Point 6: Line 190: Sensory evaluation is not correct. Why you have used 10 as the highest score? It should be an odd number (9, 7 or 5).
Response 6: Thank you very much for the constructive suggestions. We have reorganized the data and charts of the sensory evaluation, and the highest score has been set at 5. Please refer to Lines 195-196 and Figure 6.
Point 7: Line 242: filamentous structure?! It is a flat sheet with a crack.
Response 7: Thank you very much for your careful review. The mistake has been corrected in the revised manuscript. Please refer to Lines 13, 245-251.
Point 8: Figure 1: Please explain what are the right hand and the left hand images.
Response 8: Thank you very much for your careful review. We have supplemented the details about the figure caption. Please refer to Figure 2.
Point 9: Table 1: The caption and the treatments are not written.
Response 9: We are very sorry for our carelessness. We have reorganized the table in the manuscript. Please refer to Table 2, Lines 346-349.
Point 10: Sensory analysis and color parameters should be added to the manuscript (not in the supplementary file).
Response 10: Thank you very much for the constructive suggestions. Sensory analysis and color parameters have been added to the manuscript. Please refer to Figure 6, Lines 433-435

Reviewer 2 Report
Very interesting and well-designed manuscript. Just needed to be checked for minor mistakes/typos about the language.
Author Response
General Comments: Very interesting and well-designed manuscript. Just needed to be checked for minor mistakes/typos about the language.
Response to comment: Thank you for your great support for acceptance our manuscript. We have considered your comments very carefully and revisions are amended in the revised manuscript.

Reviewer 3 Report
The paper "Soluble dietary fiber extracted from grapefruit insoluble dietary fiber: physicochemical characteristics, application in blueberry jam" is well written and easy to read. The topic is interesting and even more so in the current context where it is necessary to look for other alternatives to improve the nutritional value of food products.
I will allow myself to point out some observations about the manuscript.
* I consider that it is important to include the results of color evaluation because it is difficult to understand their analysis without data or images.
* The information indicated in line 313 “Simultaneously, BJ-1 was significantly darker than BJ-2. One of the reasons might be that dietary fiber could increase the reflectance of jam, thus significantly improving the brightness of jam” is incorrect, as the BJ-2 is a sample with pectin and not dietary fiber. Revise.
* Equation (5) is not shown in Table S3, this equation is between lines 335 and 336. Check
* I consider that equations (5) and (6) should be included in the methodology and not in the results.
* Verify the acronym used in the results of the application of dietary fiber in the jam, since GPSJ-1, GPSJ-2, BJ-1 y BJ-2 were used in color, while in figure 5, where shows the behavior of the rheology properties, the following abbreviations are used: control, LM, SDF, LM+SDF. And additionally, in the text lines 342 to 351, other acronyms were used.
* Table 1 does not indicate to which samples the columns correspond. For this reason, it is not possible to verify if the analysis of the information on lines 342 to 351 is correct.
* In Figures 6 and 7, revise the acronyms to match with the information indicated on lines 358 to 371 and 374 to 388, respectively.
* I suggest that the four graphs associated with the rheological properties be in one figure with a, b, c, and d. Or check if Figures 5a and b should be better in the complementary material since they do not provide much information and also that the information presented in table 1 is obtained from these graphs.
* It is important to explain this behavior “The greater the consistency index means the greater viscosity and the stronger gel properties of the jam. BJ-2 and GPSJ-2 had higher viscosity and gel properties. Based on these, GP-IDF-SDF might have greater effect on improving gel properties of jam, but it was weaker than the effect of LM pectin”. Why does dietary fiber reduce viscosity?
* What is the typical value of rheological and textural properties of commercial jams since in the conclusion (line 396 to 399) it is indicated that the samples with dietary fiber are more stable and it is not clear what the criteria is, only that the jam with pectin have higher values is bad or not appropriate for this product? Check the analysis and conclusion
* In the sensory analysis, the behavior “Notably, the color parameters of BJ-2 and GPSJ-1 jams received higher scores due to their brighter colors” is similar to that reported with the color measurement with spectrometry. It is important to make this correlation.
* Line 408 indicates GPJ-1, and it is incorrect, the name is GPSJ-1
* I consider that the inclusion of the figure of sensory analysis should be reviewed since this information is important to validate the commercial possibility of including the new fiber in a jam.
* When you affirm “the rheology and texture data of GPSJ-1 were not outstanding, the panelists were highly receptive to its status”, you could not associate yourself with commercial products and regulations for these typical products.
Author Response
General Comments: The paper "Soluble dietary fiber extracted from grapefruit insoluble dietary fiber: physicochemical characteristics, application in blueberry jam" is well written and easy to read. The topic is interesting and even more so in the current context where it is necessary to look for other alternatives to improve the nutritional value of food products.
Response to comment: Thank you for your great support for acceptance our manuscript. We have considered your comments very carefully and revisions are amended in the revised manuscript.
Point 1: I consider that it is important to include the results of color evaluation because it is difficult to understand their analysis without data or images.
Response 1: Thank you very much for the constructive suggestions. Data of color parameters has been added to the manuscript. Please refer to Table 1, Lines 346-349.
Point 2: The information indicated in line 313 “Simultaneously, BJ-1 was significantly darker than BJ-2. One of the reasons might be that dietary fiber could increase the reflectance of jam, thus significantly improving the brightness of jam” is incorrect, as the BJ-2 is a sample with pectin and not dietary fiber. Revise.
Response 2: Thank you very much for your careful review. The mistake has been corrected in the revised manuscript. Please refer to Lines 318-326 marked in red.
Point 3: Equation (5) is not shown in Table S3, this equation is between lines 335 and 336. Check
Response 3: Thank you very much for your careful review. The mistake has been corrected in the revised manuscript. Please refer to Lines 164-166 marked in red.
Point 4: I consider that equations (5) and (6) should be included in the methodology and not in the results.
Response 4: Thank you very much for the constructive suggestions. Equations (5) and (6) have been included in the methodology. Please refer to Lines 164-166 marked in red.
Point 5: Verify the acronym used in the results of the application of dietary fiber in the jam, since GPSJ-1, GPSJ-2, BJ-1 y BJ-2 were used in color, while in figure 5, where shows the behavior of the rheology properties, the following abbreviations are used: control, LM, SDF, LM+SDF. And additionally, in the text lines 342 to 351, other acronyms were used.
Response 5: We apologize for the confusion in your review. We have standardized the abbreviations of blueberry jam (BJ, LMJ, GPSJ-1 and GPSJ-2) throughout the manuscript.
Point 6: Table 1 does not indicate to which samples the columns correspond. For this reason, it is not possible to verify if the analysis of the information on lines 342 to 351 is correct.
Response 6: We are very sorry for our carelessness. We have reorganized the table in the manuscript. Please refer to Table 2, Lines 346-349.
Point 7: In Figures 6 and 7, revise the acronyms to match with the information indicated on lines 358 to 371 and 374 to 388, respectively.
Response 7: Thank you very much for your careful review. We have standardized the abbreviations of blueberry jam (BJ, LMJ, GPSJ-1 and GPSJ-2) throughout the manuscript.
Point 8: I suggest that the four graphs associated with the rheological properties be in one figure with a, b, c, and d. Or check if Figures 5a and b should be better in the complementary material since they do not provide much information and also that the information presented in table 1 is obtained from these graphs.
Response 8: Thank you very much for the constructive suggestions. Four graphs associated with the rheological properties have been reorganized in one figure. Please refer to Figure 5, Lines 402-405.
Point 9: It is important to explain this behavior “The greater the consistency index means the greater viscosity and the stronger gel properties of the jam. BJ-2 and GPSJ-2 had higher viscosity and gel properties. Based on these, GP-IDF-SDF might have greater effect on improving gel properties of jam, but it was weaker than the effect of LM pectin”. Why does dietary fiber reduce viscosity?
Response 9: Thank you very much for your careful review. Dietary fiber could enhance the gel properties of jam, but the enhancement was not as effective as LM pectin. To avoid misunderstandings, we have rewritten this part. Please refer to Lines 363-369 marked in red.
Point 10: What is the typical value of rheological and textural properties of commercial jams since in the conclusion (line 396 to 399) it is indicated that the samples with dietary fiber are more stable and it is not clear what the criteria is, only that the jam with pectin have higher values is bad or not appropriate for this product? Check the analysis and conclusion.
Response 10: Thank you very much for the constructive suggestions. Commercial jam was not involved in our study. We prepared blueberry jam with a formulation of commercial jam, and commercial LM pectin was used to prepare LMJ. Besides, for better understanding, we have supplemented the discussions on the rheological and textural properties of blueberry jams. Please refer to Lines 406-416.
Point 11: In the sensory analysis, the behavior “Notably, the color parameters of BJ-2 and GPSJ-1 jams received higher scores due to their brighter colors” is similar to that reported with the color measurement with spectrometry. It is important to make this correlation.
Response 11: Thank you very much for the constructive suggestions. The correlation has been established between color measurement by spectroscopy and sensory analysis. Please refer to Lines 424-428 marked in red.
Point 12: Line 408 indicates GPJ-1, and it is incorrect, the name is GPSJ-1.
Response 12: We were very sorry for our carelessness, and the mistake has been corrected in the revised manuscript.
Point 13: I consider that the inclusion of the figure of sensory analysis should be reviewed since this information is important to validate the commercial possibility of including the new fiber in a jam. Response 13: Thank you very much for the constructive suggestions. Sensory analysis has been added to the manuscript. Please refer to Figure 6, Lines 433-435.
Point 14: When you affirm “the rheology and texture data of GPSJ-1 were not outstanding, the panelists were highly receptive to its status”, you could not associate yourself with commercial products and regulations for these typical products.
Response 14: Thank you very much for the constructive suggestions. To avoid misunderstandings, we have rewritten this part. Please refer to Lines 428-433.
